# Functional Implications of Estrogen and Progesterone Receptors Expression in Adenomyosis, Potential Targets for Endocrinological Therapy

**DOI:** 10.3390/jcm11154407

**Published:** 2022-07-28

**Authors:** Maria Sztachelska, Donata Ponikwicka-Tyszko, Lydia Martínez-Rodrigo, Piotr Bernaczyk, Ewelina Palak, Weronika Półchłopek, Tomasz Bielawski, Sławomir Wołczyński

**Affiliations:** 1Department of Biology and Pathology of Human Reproduction, Institute of Animal Reproduction and Food Research, Polish Academy of Sciences, 10-748 Olsztyn, Poland; d.ponikwicka-tyszko@pan.olsztyn.pl (D.P.-T.); martinezlydia998@gmail.com (L.M.-R.); e.palak@pan.olsztyn.pl (E.P.); t.bielawski@pan.olsztyn.pl (T.B.); endorepro@umb.edu.pl (S.W.); 2Departamento di Biología, Facultad de Ciencias, Universidad Autónoma de Madrid, Ciudad Universitaria de Cantoblanco, 28049 Madrid, Spain; 3Department of Pathomorphology, Medical University of Bialystok, 15-269 Białystok, Poland; piotr.bernaczyk@umb.edu.pl; 4Department of Reproduction and Gynecological Endocrinology, Medical University of Bialystok, 15-276 Białystok, Poland; wpolchlopek1@student.umb.edu.pl

**Keywords:** adenomyosis, sex steroids, estrogen receptors, progesterone receptors, aromatase, prolactin

## Abstract

Adenomyosis is a common gynaecological disease associated with the presence of endometrial lesions in the uterine myometrium. Estrogens have been proven to be the crucial hormones driving the growth of adenomyosis. Little is known about the distinct mechanisms of progesterone action in adenomyosis. Hence, in this study, we decided to characterize the expression of all nuclear and membrane estrogen and progesterone receptors. Additionally, as a functional investigation, we monitored prolactin production and cell proliferation after estradiol and progesterone treatments. We confirmed the presence of all nuclear and membrane estrogen and progesterone receptors in adenomyotic lesions at gene and protein levels. The expression of membrane progesterone receptors α and β (mPRα, mPRβ) as well as estrogen receptor β (ERβ) was upregulated in adenomyosis compared to normal myometrium. Estradiol significantly increased adenomyotic cell proliferation. Progesterone and cAMP upregulated prolactin secretion in adenomyosis in the same pattern as in the normal endometrium. In the present study, we showed the functional link between estradiol action and adenomyotic cell proliferation, as well as progesterone and prolactin production. Our findings provide novel insights into the sex steroid receptor expression pattern and potential regulated pathways in adenomyosis, suggesting that all receptors play an important role in adenomyosis pathophysiology.

## 1. Introduction

Adenomyosis is a benign uterine disease characterized by the abnormal presence of invasive endometrial glands and stromal lesions in the myometrium, surrounded by the smooth-muscle cell hyperplasia. The main symptoms of adenomyosis are dysmenorrhea, dyspareunia, uterus enlargement, heavy menstrual bleeding, and pelvic pain [1]. Adenomyosis can also be asymptomatic and related to seemingly idiopathic infertility and miscarriages.

The pathophysiology of adenomyosis remains unclear. According to the most popular invagination [2] theory, adenomyosis is a result of the invasion of endometrial tissues through the damaged junctional zone between the endometrial layer and myometrium. It has also been proposed that adenomyosis might be caused by continuous microtraumas of the myometrium that occur as a result of increased uterine peristaltic activity [3]. Other pathogenic theories state that adenomyosis results from retrograde menstruation, genetic-epigenetic mutations of endometrial cells, or metaplastic changes in stem cells from the intramyometrial embryonic pluripotential Mullerian remnants [4].

Many studies have shown that inflammatory, angiogenic, growth, and hormonal factors may play a major role in the development of adenomyosis [5]. Estrogens have been proven to be the crucial hormones driving the growth of adenomyosis. Local hyperestrogenism caused by elevated aromatase activity with normal serum levels have been shown in patients with adenomyosis [6]. Endometrial stromal cells of patients with adenomyosis have shown greater migratory and proliferative capacity, as well as a lower less apoptosis level than the endometrium from healthy controls [1,7]. Increased uterine peristaltic activity is also a result of estradiol-dependent oxytocin secretion in the myometrium [8]. Polymorphism and increased expression of estrogen receptor α (ERα) and decreased expression of progesterone receptors isoform B (PGR-B) have been connected with a higher risk of developing adenomyosis [9]. Most studies have focused on estrogen action, yet little is known about the role of progesterone in adenomyosis. In the normal endometrium, during the secretory phase of the cycle, progesterone promotes prolactin (PRL) production, the marker of decidualization and endometrial receptivity [10]. Prolactin is also responsible for the positive regulation of cell proliferation and has a protective effect against apoptosis in many types of women’s cancers [11].

Due to the lack of precise knowledge regarding adenomyosis pathomechanisms, there are no specified guidelines for diagnosis and the treatment. The pharmacological treatment of adenomyosis is focused mainly on hormonal analogues, GnRH agonists and antagonists, synthetic hormones, progestins, or aromatase inhibitors [6,12]. Despite so many treatment options available, pharmacotherapy is predominantly futile. The most common and effective treatment is hysterectomy [13]. In younger patients with the need for fertility preservation, e.g., uterus-sparing resection or artery embolization can be performed [14]. Still, even after surgical removal of the adenomyotic lesions, relapse occurs frequently.

Hence, in the present study, we decided to characterize the expression profile of estrogen and progesterone receptors in order to identify new molecular targets for potential adenomyosis pharmacotherapy. Additionally, we monitored prolactin production after estradiol and progesterone treatments in adenomyosis, as a functional investigation of potentially activated pathways.

## 2. Materials and Methods

### 2.1. Tissues Collection

Adenomyosis (*n* = 30) and myometrium (*n* = 30) tissues were collected during laparoscopy from the patients of the Department of Reproduction and Gynecological Endocrinology at the Medical University of Bialystok, Poland. The diagnosis of adenomyosis was performed based on the patients’ medical history and interview, as well as ultrasound and laparoscopic examination, then later confirmed by the histopathological examination of each lesion. Patients diagnosed with co-occurring endometriosis were excluded from the study.

Endometrial (*n* = 10) samples were obtained by aspiration biopsy from women undergoing IVF treatment for male factor infertility in the Department of Reproduction and Gynecological Endocrinology at the Medical University of Bialystok, Poland. Biopsies were collected from healthy, ovulating, and regularly menstruating women aged 25–40. Biopsies were performed in the second half of the natural cycle (7 days after ovulation) that preceded the IVF cycle, employing Pipelle biopsy catheter (CCD, Paris, France)

All the tissues were preserved for further IHC (in 4% formalin) and gene expression analysis (snap frozen and stored in −80 °C) and cryo-preserved for cell isolation and tissue explant stimulations. The Human Investigation Ethics Committees at the Medical University of Bialystok approved the study. Written informed consent was obtained from all the patients.

### 2.2. Total RNA Isolation

The total RNA was isolated from the homogenated tissues using the TRIzol extraction method (Invitrogen, Carlsbad, CA, USA). The purification with 75% ethanol was repeated thrice in order to obtain better purity of the isolated RNA. The quantity and quality of total RNA were checked spectrophotometrically using the NanoDrop ND-1000 (NanoDrop Technologies, Wilmington, DE, USA) and electrophoretically on an agarose gel.

### 2.3. Reverse Transcription and Real-Time PCR

Briefly, 1 µg of the isolated RNA was treated with DNase I (Thermo Fisher Scientific, Waltham, MA, USA) to remove any possible DNA contamination. The reverse transcription reaction was performed with the High Capacity cDNA Reverse Transcription Kit (Thermo Fisher Scientific, Waltham, MA, USA) according to the manufacturer’s instructions. The quantitative polymerase chain reaction was performed with the SYBR Green qPCR Master Mix (Thermo Fisher Scientific, Waltham, MA, USA) and a Step One Plus Real-Time PCR System thermal cycler (Applied Biosystems, Foster City, CA, USA). The PCR products were checked by melting curve analysis and by agarose gel electrophoresis. Gene expression was normalized in relation to the beta-actin gene (*ACTB*). Primers sequences are listed in Table A1 (Appendix A).

### 2.4. Immunohistochemistry

Formalin-fixed paraffin sections of adenomyosis and myometrial tissues were placed on microscope slides, hydrated, and dewaxed in xylene and a series of decreasing alcohol concentrations. Slides were incubated for 20 min in 10 mM citrate buffer pH 6.0 in a steam autoclave (121 °C). Then, the slides were cooled to room temperature (RT) and incubated for 1 h in blocking buffer containing 3% BSA (Sigma-Aldrich, Saint Louis, MO, USA) in PBS with 0.05% Tween 20 at room temperature, followed by the addition of anti-ERα (M7047, Dako, Glostrup, Denmark; dilution 1:200), anti-ERβ (MCA1974, AbD Serotec, Oxford, UK; dilution 1:1000), anti-aromatase (sc-30086, Santa Cruz Biotechnology, Dallas, US; dilution 1:2000), anti-PGR (8757S, Cell Signaling Technology, Danvers, MA, USA; dilution 1:500), anti-PGRMC1 (13856S, Cell Signaling Technology, Danvers, MA, USA 1:500), anti-PGRMC2 (H00010424-M04, Abnova, Taipei, Taiwan; 1:1000 dilution), anti-mPRα (ab75508, Abcam, Cambridge, UK; 1:500 dilution), anti-mPRβ (ab46535, Abcam, Cambridge, UK; 1:1000 dilution), and anti-mPRγ (ab79517, Abcam, Cambridge, UK; 1:500 dilution) antibodies and incubated overnight at 4 °C. Endogenous peroxidase activity was blocked with 0.5% H_2_O_2_ in PBS for 20 min at RT. The sections were then incubated with the anti-mouse or anti-rabbit Envision^®^ + System-HRP polymer (Dako, Glostrup, Denmark) for 30 min at room temperature. The reaction products were visualized using 3′-3-diaminobenzidine tetrahydrochloride (DAB, Dako, Glostrup, Denmark). Hematoxylin was used as a contrast dye. Sections were dehydrated in an alcohol series with increasing concentration and secured with a coverslip using Pertex formulation adhesive (Histolab Products AB, Askim, Sweden).

### 2.5. Quantification of IHC Stainings in Myometrium and Adenomyosis Tissues

The staining intensity was measured as the optical density of the defined areas on the stained microscope slides imaged by ImageJ [15]. For every investigated protein, six slides were included in the analysis. Six areas were randomly selected from each section and automatically quantified. For each analyzed section, an average optical density was calculated. Summarized results are presented in Appendix A (Figure A1).

### 2.6. Primary Cells Isolation

Adenomyotic tissues were thawed and cut into small pieces in a glass dish, then treated with 1% collagenase solution (Sigma-Aldrich, Saint Louis, MO, USA) dissolved in DMEM/F-12 culture medium (Gibco, Paisley, UK). Incubation was carried out at 37 °C in the presence of 5% CO_2_, mixing the suspension with a serological pipette every 15 min. In order to block the enzymatic activity of collagenase, 5 mL of culture medium was added, containing DMEM/f-12 (Gibco, Paisley, UK) with the addition of 10% bovine serum (FBS, Biochrom, Berlin, Germany) and Antibiotic Antimycotic Solution (Sigma-Aldrich, Saint Louis, MO, USA). To remove undigested tissue fragments and cell aggregates, the cell suspension was sieved through a 40-μm sieve, then centrifuged for 10 min at 300× *g*. The cell pellet was suspended in 2 mL of DMEM/f-12 culture medium (Gibco, Paisley, UK) with the addition of 10% FBS (Biochrom, Berlin, Germany) and Antibiotic Antimycotic Solution (Sigma-Aldrich, Saint Louis, MO, USA), then transferred to a 75 cm^2^ cell culture flask. The medium was supplemented to 10 mL and changed after 45–60 min with fresh medium in order to remove the remaining epithelial cells, leukocytes, and erythrocytes. Cultures were carried out for 3–4 days at 37 °C in a humid atmosphere in the presence of 5% CO_2_, with the substrate changed every two days until reaching 90% confluence. Subsequently, cells were trypsinized (0.25%, Gibco, Paisley, UK) and seeded into 96-well plates for 72 h for estradiol and progesterone stimulations and proliferation assessment. Adenomyotic cell cultures purity was confirmed by anti-vimentin staining following standard immunofluorescence staining procedures.

### 2.7. Cell Proliferation Assay

The cell proliferation assay was evaluated in adenomyosis primary cell lines using BrdU Cell Proliferation Assay kit (#6813, Cell Signaling, Danvers, MA, USA) and performed following the manufacturer’s instructions. Cells were treated with either vehicle, estradiol (Sigma-Aldrich, Saint Louis, MO, USA; 1, 10, 30, 100 nM), or progesterone (Sigma-Aldrich, Saint Louis, MO, USA; 0.01, 0.1, 1 μM). Absorbance was read at 450 nm using the TECAN Tecan Infinite M200pro microplate reader (Tecan, Reading, UK).

### 2.8. Explants Stimulations

Endometrial and adenomyotic tissues were thawed and washed with PBS and cut with a sterile scalpel into 1 mm cube pieces on a sterile glass Petri dish. Then, tissues explants were plated on 24-well plates and incubated for 24 h in basal culture medium (DMEM/f-12 (Gibco, Paisley, UK) with 10% FBS (Gibco, Paisley, UK) and 1% Antibiotic Antimycotic Solution (Sigma-Aldrich, Saint Louis, MO, USA), at 37 °C and in humidified atmosphere with 5% CO_2_ concentration, and then for 6 h in stimulation medium (DMEM/f-12 with 0.5% depleted FBS (Biochrom, Berlin, Germany) and 1% Antobiotic Antimycotic solution). After this step, tissues were exposed either to vehicle, estradiol 30 nM, progesterone 1 μM, cAMP 100 μM, or combined. After 72 h, media were collected for further analysis. All of the stimulants were provided by Sigma-Aldrich (Saint Louis, MO, USA. Three independent experiments, in quadruplicate, were performed for each treatment. Every repetition of each experiment was prepared using the explant derived from one patient.

### 2.9. Prolactin Secretion Measurements

Culture media collected from the explant stimulations were used for the evaluation of PRL secretion. PRL secretion was measured in the endometrial and adenomyosis explants media with Prolactin Kit (Roche, Tokyo, Japan) according to manufacturer’s protocol using Cobas e411 analyzer (Roche Diagnostic Ltd., Basel, Switzerland).

### 2.10. Statistics

Statistical significance was calculated with one-way ANOVA with the post hoc Bonferroni test using GraphPad PRISM v. 7.0 (GraphPad Software, Inc., La Jolla, CA, USA). Results with *p* ≤ 0.05 were considered statistically significant. All the results are given as mean ± SEM.

## 3. Results

### 3.1. Estrogen and Progesterone Receptors Are Expressed in Adenomyosis and Normal Myometrium

qPCR analysis demonstrated all progesterone and estradiol receptor expression in both normal myometrium and adenomyosis (Figure 1 and Figure 2). Nuclear progesterone receptor (PGR) and progesterone receptor membrane components 1 and 2 (PGRMC1 and PGRMC2) showed the highest expression level among all progesterone receptors (Figure 1a,i,k). The expression of PAQR7 (mPRα) and PAQR8 (mPRβ) was significantly upregulated in adenomyosis compared to normal myometrium (Figure 1c,e). No significant upregulation of *PAQR5* was observed in adenomyosis compared to normal myometrium (Figure 1g).

Immunohistochemistry analysis confirmed the presence of all the progesterone receptors in adenomyosis and normal myometrium (Figure 1 and Figure A1). Strong nuclear-cytoplasmic expression of the PGR was localized both in the stromal cells and in the glandular epithelium of adenomyosis. The expression of membrane progesterone receptor α (mPRα) was localized mainly in the cytoplasm of the adenomyotic glandular tissue (Figure 1d). Membrane progesterone receptor β (mPRβ) was localized both in the stromal cells and the glandular epithelium of adenomyosis (Figure 1f). Expression of the membrane progesterone receptor γ (mPRγ) receptor was localized in the cytoplasm of the glandular epithelium (Figure 1h). PGRMC1 receptor showed a strong cytoplasmic expression in the adenomyotic glandular epithelium and the stromal cells (Figure 1j). PGRMC2 also localized in the cytoplasm of the glandular epithelium and the stromal cells (Figure 1). The protein expression of PGR, mPRα, mPRβ, and PGRMC1 was significantly higher in adenomyosis compared to the normal myometrium (Figure A1a–c,e), while mPRγ and PGRMC2 had similar expression in both tissues (Figure A1d,f).

All estradiol receptor (ESR1, ESR2, GPER) expression levels were significantly higher in adenomyosis than in normal myometrium (Figure 2a,c,e). ESR2 expression in myometrium was barely detectable (Figure 2c). Immunolocalization showed strong expression of estrogen receptor α (Erα) and estrogen receptor β (Erβ) in the nucleus of adenomyotic glandular epithelium and stromal cells (Figure 2b,d). No expression of ERβ was observed in the myometrium (Figure 2d and Figure A1h). GPER expression was detected in the cytoplasm of the glandular epithelium and the stromal cells of adenomyosis (Figure 2f). Cytoplasmic localization of GPER in the myometrium was also observed (Figure 2f). Protein expression of ERα and GPER was significantly higher in adenomyosis compared to the normal myometrium (Figure A1g,i).

### 3.2. Aromatase Is Expressed Both in Adenomyosis and in the Normal Myometrium

Significantly higher *CYP19A1* expression in adenomyosis than in normal myometrium at both the gene and protein levels was observed (Figure 3a and Figure A1j). Immunohistochemical localization showed the abundant presence of aromatase in the glandular epithelium of adenomyosis (Figure 3b). Weak staining was also observed in the stromal cells of adenomyosis (Figure 3b).

### 3.3. Adenomyotic Cell Proliferation Is Promoted by Physiological Levels of Estradiol

Adenomyotic cell proliferation was significantly increased after stimulation with 10 or 30 nM of estradiol (Figure 4a). Other concentrations of estradiol (1 and 100 nM) did not affect adenomyosis cell proliferation (Figure 4a). Similarly, none the concentrations of progesterone (0.01–1 µM) had an effect on adenomyotic cell proliferation (Figure 4b).

### 3.4. Prolactin Is Expressed and Secreted by Adenomyotic Tissue

The expression of prolactin (*PRL*) and prolactin receptor (*PRLR*) was significantly upregulated in adenomyosis compared to normal myometrium (Figure 5a,b). cAMP, progesterone with cAMP, and estradiol with progesterone and cAMP increased prolactin secretion in adenomyotic explants (Figure 6). Progesterone alone and progesterone with estradiol had an additive effect to the cAMP upregulation of prolactin production. Furthermore, the same stimuli increased prolactin production in the endometrium, which was used as a positive control for in vitro prolactin production (Figure 6).

## 4. Discussion

Due to the lack of knowledge pertaining to the molecular bases of the pathogenesis of adenomyosis currently available, diagnostic and treatment options are limited [16,17,18]. Recent studies have shown similarities with deep endometriosis at the histological level [19]. Local hyperestrogenism and aromatase activity in adenomyotic lesions was also demonstrated [20]. Yet, little is known about the distinct mechanisms of estrogen and progesterone receptors action in adenomyosis. We observed significantly higher expression of all three types of estrogen receptors in adenomyosis compared to the normal myometrium. ERα and ERβ were localized in both epithelial and stromal cells of adenomyosis. An analogous pattern of expression has previously been demonstrated in endometriotic tissues [21,22]. We also showed upregulated ERβ expression in adenomyotic lesions. It has been suggested that ERβ overexpression could be responsible for inflammation in adenomyosis [23]. On the contrary to previous studies, we did not find ERβ expression in normal myometrium [24,25]. It has been indicated that estrogens through ERα can promote the proliferation of endometrium or human epithelial cell line [26]. Study on a mice model has shown that ERα can carry out its effects on endometrial epithelial cell proliferation independent of classical genomic signaling, suggesting non-classical signaling pathway involvement [27]. We also showed that estradiol in physiological concentrations (10, 30 nM) significantly stimulated adenomyotic cell proliferation. Higher concentrations of estradiol did not promote proliferation, as expected [28]. However, more studies are needed to identify the key estrogen receptor involved in regulation of cell proliferation in adenomyosis. Besides genomic signaling, estrogen may exert its effects through activation of the non-classical GPER receptor [29]. We demonstrated high gene and protein GPER expression in both adenomyosis and normal myometrium tissues. GPER overexpression has also been observed in eutopic and ectopic endometrium of patients with endometriosis [30]. This unique GPER expression pattern in endometriosis as compared to the normal endometrium suggests a potential role for GPER in the hormonal regulation of endometriosis [31]. Recent studies have also shown that GPER activation enhanced contractile responses to oxytocin in the myometrium [32]. This finding may support the continuous microtraumas theory of adenomyosis development. Aromatase is the maim enzyme involved in estradiol synthesis. It enables estradiol production in adenomyosis and myometrial cells [33]. Aromatase activity in the adenomyotic lesions, similar to endometriosis [21], was higher than that in normal endometrium or myometrium [24,34]. We also detected aromatase gene and protein expression in both tissues. It has been suggested that the upregulation of aromatase and ERβ genes expression might represent a potential risk factor for adenomyosis [35] and endometriosis development [36]. Moreover, upregulation of aromatase was positively correlated with the presence of dysmenorrhea and infertility regardless of endometriosis occurrence [37].

Progesterone resistance resulting from the decreased expression of PGR, especially PGR-B isoform, has been reported in adenomyosis [38]. We showed PGR expression at both gene and protein levels in adenomyotic tissue. The effect of progesterone has been shown to be dependent on the proportion and localization of the receptor isoforms in the cell. The different cellular localization of the isoforms, PGR-B outside the nucleus and PGR-A inside the nucleus, may account for isoform-specific activity in the cell [39,40]. It has been shown that PGR-B can promote uterine epithelial cell proliferation, but only when it is not repressed by PGR-A [41]. PGR-B was reported to be suppressed by DNA methylation in adenomyotic stromal cells [42]. In our study, progesterone did not stimulate adenomyotic cell proliferation, which seems to be related to PGR-B repression in adenomyosis. The importance of PGR in adenomyosis has been confirmed in recent study, where significant relief from dysmenorrhea was observed after the administration of selective progesterone receptor modulator mifepristone [43].

In contrast to endometrial cancer [44] and endometriotic tissues [45] we did not observe decreased expression of progesterone membrane receptors in adenomyosis. Previous studies have shown that the mPRα and mPRβ expression at gene as well as protein levels was significantly downregulated in the ectopic endometrium of patients with endometriosis compared to the endometrium of healthy women [45]. We detected strong expression of mPRβ both in the stromal cells and the glandular epithelium, as well as mPRα and mPRγ, mainly in the cytoplasm of the glandular tissue, suggesting that adenomyosis is a separate disease, and may express molecular differences compared to endometriosis. Progesterone receptor membrane components PGRMC1 and PGRMC2 also showed strong cytoplasmic expression in the glandular epithelium and the stromal cells of adenomyotic lesions. Decreased expression of PGRMC1 and PGRMC2 has been reported in the eutopic endometrium of patients with endometriosis compared to the endometrium of healthy women [46]. Furthermore, decreased expression level of PGRMC2 may be related to the progesterone insensitivity often observed in the endometrium of non-human primates with endometriosis [47].

It is well known that progesterone stimulates the endometrium to directly synthesize prolactin [48,49]. We detected the upregulation of prolactin secretion in adenomyosis in the same pattern as in the normal endometrium, mediated by both cAMP and progesterone. It has been suggested that prolactin may be involved in the pathogenesis of benign uterine diseases, including adenomyosis and endometriosis [50]. Moreover, studies in mice models have shown the induction of adenomyosis with intrauterine pituitary isografts [51,52]. Similar to our findings, the overexpression of PRL-R has been found in bovine models of uterine adenomyosis [53]. High doses of prolactin have been proven to have an antiapoptotic effect in certain types of tumors, e.g., breast cancer or myeloma, and support their growth potency [54,55]). However, only local prolactinemia seems to play a role in adenomyosis, as no correlation between prolactin levels in serum and adenomyosis incidence has been found [56]. Further investigations are necessary to characterize the signaling pathways underlying the local prolactin production in adenomyosis.

## 5. Conclusions

The pathophysiology of adenomyosis is complex and still not well understood. Thus, a better characterization of the receptors and pathways activated in adenomyosis may identify new molecular targets for potential adenomyosis pharmacotherapy. In the present study, we characterized the expression profile of all estrogen and progesterone receptors and evaluated estradiol and progesterone role in the regulation of cell proliferation and prolactin production. We showed the functional link between estradiol action and adenomyotic cell proliferation as well as progesterone and prolactin production. Our findings provide novel insights into the sex steroid receptor expression pattern and potential regulated pathways in adenomyosis, suggesting that all receptors play an important role in adenomyosis pathophysiology.

## Figures and Tables

**Figure 1 jcm-11-04407-f001:**
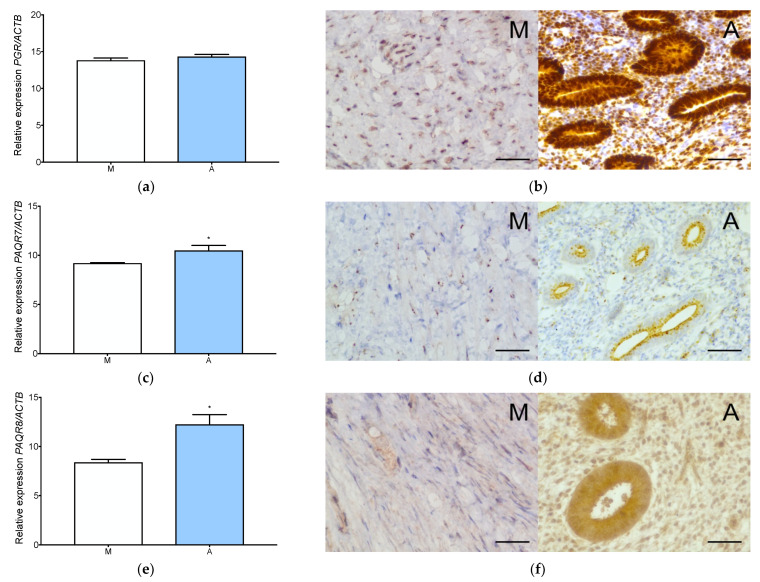
Progesterone receptors expression in adenomyosis and normal myometrium. Characterization of *PGR* expression at gene (**a**) and protein (**b**) level, *PAQR7* gene (**c**) and protein (**d**) level, *PAQR8* gene (**e**) and protein (**f**) level, *PAQR5,* gene (**g**) and protein (**h**) level, *PGRMC1* gene (**i**) and protein (**j**) level and *PGRMC2* gene (**k**) and protein (**l**) level. The columns represent the ratio of the expression of the gene tested and *ACTB* ± SEM. Asterisk indicates significant differences (* *p* ≤ 0.05). Original magnification, 40×; scale bar, 50 μm. A, adenomyosis; M, myometrium.

**Figure 2 jcm-11-04407-f002:**
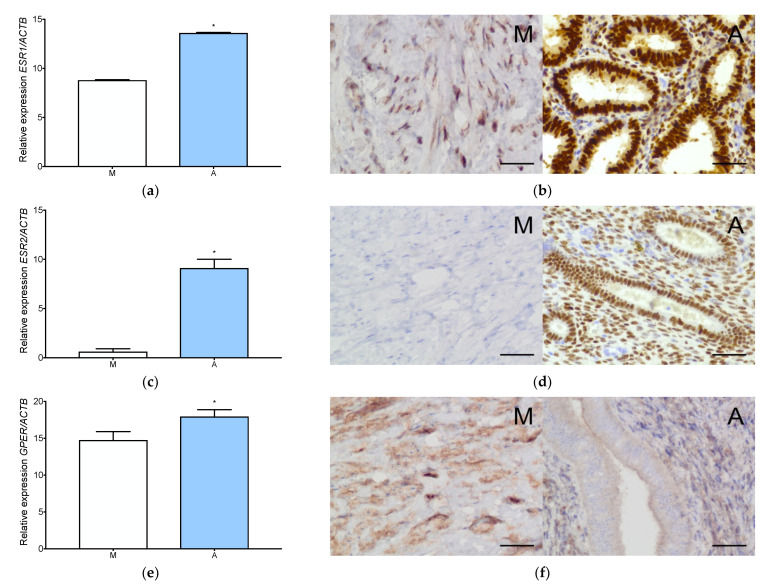
Estrogen receptors expression in adenomyosis and normal myometrium. Characterization of ESR1 expression at gene (**a**) and protein (**b**) level, ESR2 expression at gene (**c**) and protein (**d**) level and GPER expression at gene (**e**) and protein (**f**) level in myometrium and adenomyosis. Each column represents the ratio of the expression of investigated gene and *ACTB* ± SEM. Asterisk indicates significant differences (* *p* ≤ 0.05). Original magnification, 40×; scale bar, 50 μm. A, adenomyosis; M, myometrium.

**Figure 3 jcm-11-04407-f003:**
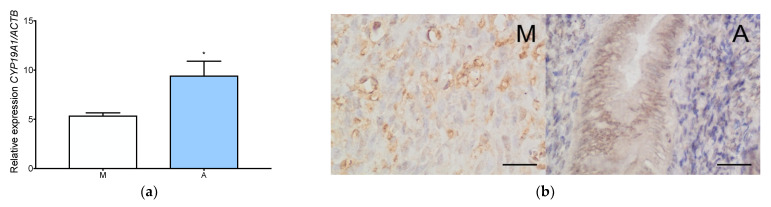
Aromatase expression in adenomyosis and normal myometrium. RT-qPCR analysis of CYP19A1 expression level in myometrium and adenomyosis (**a**). Each column represents the ratio of the expression of investigated gene and *ACTB* ± SEM. Asterisk indicates significant differences (* *p* ≤ 0.05). Immunohistochemical localization of aromatase in adenomyosis and normal myometrium (**b**). Original magnification, 40×; scale bar, 50 μm. A, adenomyosis; M, myometrium.

**Figure 4 jcm-11-04407-f004:**
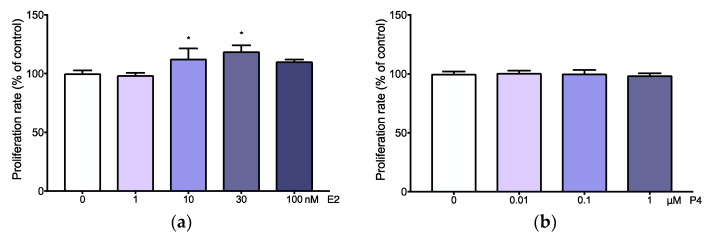
Adenomyotic cells proliferation after estradiol and progesterone stimulations. Adenomyosis cells proliferation rate after estradiol (**a**) and progesterone (**b**) treatment in different concentrations. Cell proliferation of the treated groups is presented as the percentage of the control (considered as 100%). Asterisks indicate significant differences (* *p* ≤ 0.05).

**Figure 5 jcm-11-04407-f005:**
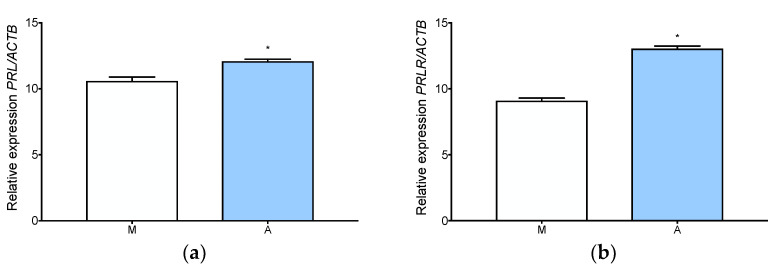
Prolactin and prolactin receptor expression in adenomyosis and normal myometrium. RT-qPCR analysis of *PRL* (**a**) and *PRLR* (**b**) expression of level in myometrium and adenomyosis. Each column represents the ratio of the expression of investigated gene and *ACTB* ± SEM. Asterisks indicate significant differences (* *p* ≤ 0.05). M, myometrium; A, adenomyosis.

**Figure 6 jcm-11-04407-f006:**
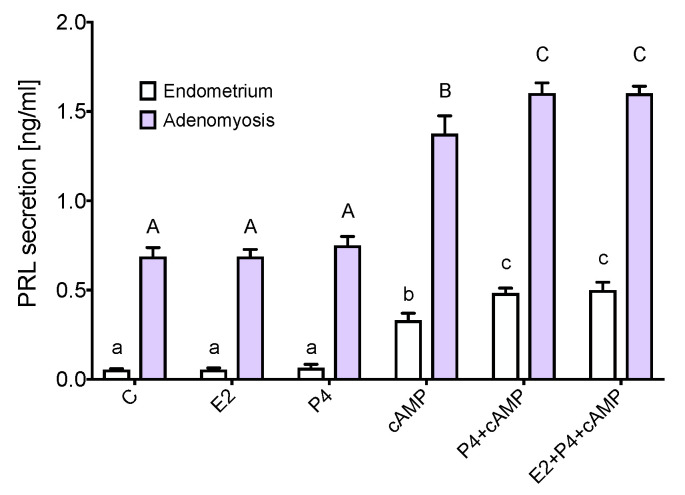
Prolactin secretion in endometrial and adenomyotic explants. Secretion of prolactin in endometrial and adenomyotic explants after 72 h stimulations. Significant differences (*p* < 0.05) are indicated by different letters above the bars (lower- and upper case letters for endometrium and adenomyosis, respectively). C, control; cAMP, 8-bromo-cAMP; E2, estradiol; M, myometrium; P4, progesterone.

## Data Availability

The data presented in this study are available on request from the corresponding author.

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
