# Peer review of "Functional Implications of Estrogen and Progesterone Receptors Expression in Adenomyosis, Potential Targets for Endocrinological Therapy"

_jcm, 2022, doi:10.3390/jcm11154407_

Round 1
Reviewer 1 Report
Point by point review to article titled: “
Functional implications of estrogen and progesterone receptors expression in adenomyosis; potential targets for endocrinological therapy.”
The aim of the work was to evaluate the expression levels of estrogen and progesterone receptors in adenomyosis and to study the functional link between estradiol and progesterone action with stromal adenomyosis phenotype and/or adenomyosis explants in culture.
The quality of the study design and methodology are suitable for the aim of the work. The visual materials are adequately presenting the results and the interpretation of the data is presented in the light of existing knowledge. However, I have some minor points needing revision, before considering the manuscript for publication.
1. In In Materials and methods paragraph (2.1.) the authors are not stating whether in adenomyosis patients the existence of endometriosis was excluded. If yes, please specify.
2. In Materials and methods paragraph (2.1 the authors state the collection of endometrial (n=10) samples from women undergoing IVF treatment for male factor infertility. Although used as positive control for prolactin production an additional control in this case would be endometrium of women with endometriosis. By comparing adenomyosis to eutopic endometrial tissue the differences in the hormonal responses between endometriosis and adenomyosis might point to putative differences between those two diseases.
3. How was the purity of the primary stroma cells cultures determined?
4. In Results section (p6) the authors are stating that:” Immunohistochemistry analysis confirmed the presence of all the progesterone receptors in adenomyosis and normal myometrium (Figure 1B). Strong nuclear-cytoplasmic expression of the PGR was localized both in the stromal cells and in the glandular epithelium of adenomyosis. ’’However, Fig 1b is showing the nuclear localization of the PGR in both epithelial and stroma cell compartments in adenomyosis and nuclear-cytoplasmic localization of the target in the myometrium. In addition, the figure 1b is only related to PGR and not to all PGR subtypes.
Based on the mRNA and protein data for PGR given in the current manuscript version the mRNA levels seem to be identical between M and A (figure 1a) but the levels of the protein are much higher in A when compared to M. Can the authors discuss this phenomenon?
5. It will be interesting for the reader to see summarized evaluation of the staining intensity of all IHC targets for all of the patients in the study given in graphical form.
6. Figure 1d and 1m: Please replace the IHC photo for A with one of better quality.
7. In Results section (p6) the authors are stating that:’’ Strong expression of estrogen receptor α (ERα) and estrogen receptor β (ERβ) is detected in the nucleus and cytoplasm of adenomyotic glandular epithelium and stromal cells (Figure 2b, 2d).’’ However as for PGR, the staining for both targets is to my opinion mainly nuclear.
Author Response
Dear Rewiever,
Thank You for Your valuable comments, please find our answers below.
- In Materials and methods paragraph (2.1.) the authors are not stating whether in adenomyosis patients the existence of endometriosis was excluded. If yes, please specify.
Patiens diagnosed with co-occurring endometriosis were excluded from the study. The diagnosis was made on the basis of patients’ medical history and interview, ultrasound and laparoscopic examination. This information was added to the manuscript (line 84).
- In Materials and methods paragraph (2.1 the authors state the collection of endometrial (n=10) samples from women undergoing IVF treatment for male factor infertility. Although used as positive control for prolactin production an additional control in this case would be endometrium of women with endometriosis. By comparing adenomyosis to eutopic endometrial tissue the differences in the hormonal responses between endometriosis and adenomyosis might point to putative differences between those two diseases.
Adenomyotic tissues were obtained during surgeries. Endometrial tissues were obtained from biopsies. Both tissues were removed as a part of medical procedures. Unfortunately, we do not have access to eutopic endometrium from patients with endometriosis. Yet the suggestion is very valuable and we will do our best to include this experiment in our future studies.
- How was the purity of the primary stroma cells cultures determined?
Conditions for cell isolation were specifically designed to obtain pure cultures of adenomyotic stromal cells. Using low concentration of collagenase with short incubation time resulted in digesting only stromal tissue, while fibrosis requires higher concentration and longer incubation time. Purity of adenomyosis stromal cell cultures was confirmed by vimentin staining.
- In Results section (p6) the authors are stating that:” Immunohistochemistry analysis confirmed the presence of all the progesterone receptors in adenomyosis and normal myometrium (Figure 1B). Strong nuclear-cytoplasmic expression of the PGR was localized both in the stromal cells and in the glandular epithelium of adenomyosis. ’’However, Fig 1b is showing the nuclear localization of the PGR in both epithelial and stroma cell compartments in adenomyosis and nuclear-cytoplasmic localization of the target in the myometrium. In addition, the figure 1b is only related to PGR and not to all PGR subtypes.
We apologize for the typing mistake that led to the misunderstanding. The sentence should state that “Immunohistochemistry analysis confirmed the presence of all the progesterone receptors in adenomyosis and normal myometrium (Figure 1)”. This has been corrected in the new version of the manuscript.
We also corrected the IHC image, as ERα was accidentally added instead of the proper PGR staining image. This change has no impact on the interpretation of the data.
Based on the mRNA and protein data for PGR given in the current manuscript version the mRNA levels seem to be identical between M and A (figure 1a) but the levels of the protein are much higher in A when compared to M. Can the authors discuss this phenomenon?
Protein expression levels may differ between tissues are the result of various mechanisms for initiation and blocking translation or post translational modifications. More experiments are needed to clarify this observation.
- It will be interesting for the reader to see summarized evaluation of the staining intensity of all IHC targets for all of the patients in the study given in graphical form.
We evaluated the staining intensity of all IHC targets with ImageJ as advised.
- Figure 1d and 1m: Please replace the IHC photo for A with one of better quality.
Figures we corrected according to the comment.
- In Results section (p6) the authors are stating that:’’ Strong expression of estrogen receptor α (ERα) and estrogen receptor β(ERβ) is detected in the nucleus and cytoplasm of adenomyotic glandular epithelium and stromal cells (Figure 2b, 2d).’’ However as for PGR, the staining for both targets is to my opinion mainly nuclear.
Thank You for the important comment, we corrected the results description accordingly.
Reviewer 2 Report
The authors focused on adenomyosis tissue and evaluated the expression of hormone receptors. Not many studies about these receptors on adenomyosis tissue have been published, so this is an interesting and valuable report, however there are several problems for publication.
1) The photographs used in Figure 1 and Figure 2 appear that each antibody was evaluated at a different site to represent the intensity of immunostaining. Is it possible to make a more objective evaluation for example, using H-score or other methods instead of using only one site of these photographs?
2) Adenomyosis tissue is invasive and has surrounding fibrosis, so it would be difficult to use only adenomyosis tissue in the primary culture. Are the authors able to prove that the cells obtained from primary culture are truly adenomyotic cells?
3) Why does the stimulating cell proliferation by E2 in Figure 4 disappear at 100nM even though it is observed at 10nM and 30nM? Please add to the discussion about this point.
Author Response
Dear Reviewer,
Thank You for Your valuable comments, please find our answers below.
1) The photographs used in Figure 1 and Figure 2 appear that each antibody was evaluated at a different site to represent the intensity of immunostaining. Is it possible to make a more objective evaluation for example, using H-score or other methods instead of using only one site of these photographs?
We evaluated the staining intensity of all IHC targets with ImageJ like advised.
2) Adenomyosis tissue is invasive and has surrounding fibrosis, so it would be difficult to use only adenomyosis tissue in the primary culture. Are the authors able to prove that the cells obtained from primary culture are truly adenomyotic cells?
Conditions for cell isolation were specifically designed to obtain pure cultures of adenomyotic stromal cells. Using low concentration of collagenase with short incubation time resulted in digesting only stromal tissue, while fibrosis requires higher concentration and longer incubation time. Purity of adenomyosis stromal cell cultures was confirmed by vimentin staining.
3) Why does the stimulating cell proliferation by E2 in Figure 4 disappear at 100nM even though it is observed at 10nM and 30nM? Please add to the discussion about this point.
Results from out former study suggested that high doses of estradiol (above 30nM) reduces cell viability (Sztachelska et al., 2019), thus 100nM should not promote proliferation. This point was added to discussion (line 283).
Round 2
Reviewer 2 Report
The authors have added some informative data in the revised edition, but some points are still unclear and require further revision.
It is still unclear whether adenomyotic cells were truly obtained by the method used for primary cell isolation. The authors used an anti-Vimentin antibody to confirm that however, Vimentin is also present in the myometrium and in surrounding fibroblast as well as adenomyosis. Also, ER and PGR appear to be strongly expressed in the epithelium, but this experimental technique seems to only collect stroma cells, and thus requires a more rigorous description about characterization of adenomyotic cells.
Author Response
Dear Reviewer,
thank You for Your insightful comment, please find our response below.
"It is still unclear whether adenomyotic cells were truly obtained by the method used for primary cell isolation. The authors used an anti-Vimentin antibody to confirm that however, Vimentin is also present in the myometrium and in surrounding fibroblast as well as adenomyosis. Also, ER and PGR appear to be strongly expressed in the epithelium, but this experimental technique seems to only collect stroma cells, and thus requires a more rigorous description about characterization of adenomyotic cells."
Our goal was to isolate stromal cells only, as they are responsible for prolactin secretion in the normal endometrium and, as we determined, adenomyotic lesions. Thus, anti-vimentin staining was performed to determinate adenomyotic stromal cell cultures purity and to exclude the possibility of epithelial cells contamination. As we applied the specific isolation protocol we do not expect the possibility of myometrial/fibroblast contamination, but to dispel any doubts we could perform additional experiment. We cryopreserved and stored the isolated adenomyotic cells in the same passage as was used in the proliferation assay, thus it is possible to thaw and culture them, and check their specificity with immunocytochemical staining with the anti-smooth muscle markers antibodies (eg. α-Smooth Muscle Actin) and anti-ERβ antibody. Negative staining for smooth muscle markers would confirm the non-myometrial origin of the cells, while positive staining for ERβ would confirm their adenomyotic provenience.